# Chemokine Receptor Expression on T Cells Is Modulated by CAFs and Chemokines Affect the Spatial Distribution of T Cells in Pancreatic Tumors

**DOI:** 10.3390/cancers14153826

**Published:** 2022-08-06

**Authors:** Laia Gorchs, Marlies Oosthoek, Tülay Yucel-Lindberg, Carlos Fernández Moro, Helen Kaipe

**Affiliations:** 1Department of Laboratory Medicine, Karolinska Institutet, 141 52 Stockholm, Sweden; 2Department of Dental Medicine, Karolinska Institutet, 141 52 Stockholm, Sweden; 3Department of Pathology and Cancer Diagnostics, Karolinska University Hospital, 141 57 Stockholm, Sweden; 4Clinical Immunology and Transfusion Medicine, Karolinska University Hospital, 141 52 Stockholm, Sweden

**Keywords:** pancreatic cancer, carcinoma-associated fibroblasts, CD4 T cells, CD8 T cells, chemokines, CXCR3, CCR5, CXCR4

## Abstract

**Simple Summary:**

The infiltration of T cells in pancreatic tumors has been correlated with better overall survival. However, the dense desmoplastic stroma, mainly composed by cancer-associated fibroblasts (CAFs), can sequester the T cells in the stroma preventing them from reaching the tumor nests. Chemokines are small molecules capable of directing T cell migration. Here, we explored whether CAFs could modulate the expression of chemokine receptors on T cells and examined if the spatial distribution of T cells within tumors was correlated to chemokine secretion patterns. Overall, we found that CXCR3 ligands was associated with an increased number of T cells in tumor rich areas and that CAFs downregulated the expression of CXCR3 on T cells. Understanding the mechanisms by which T cells are prevented from reaching the tumor nests is of great importance for the development of novel targeting therapies.

**Abstract:**

The accumulation of T cells is associated with a better prognosis in pancreatic cancer. However, the immunosuppressive tumor microenvironment, largely composed by cancer-associated fibroblasts (CAFs), can prevent T cells from reaching the tumor nests. We examined how human CAFs modulated chemokine receptors known to be associated with T cell trafficking, CXCR3 and CCR5, and T cell exclusion, CXCR4. CAFs decreased the expression of CXCR3 and CCR5 but increased CXCR4 expression in both 2D and 3D cultures, affecting the migratory capacity of T cells towards CXCL10. An immunohistochemistry analysis showed that very few T cells were found in the tumor nests. Within the stroma, CD8^+^ T cells were localized more distantly from the malignant cells whereas CD4^+^ T cells were more equally distributed. Tumor tissues with a high production of chemokines were associated with less T cell infiltration when the whole tissue was analyzed. However, when the spatial localization of CD8^+^ T cells within the tissue was taken into account, levels of CXCR3 ligands and the CCR5 ligand CCL8 showed a positive association with a high relative T cell infiltration in tumor-rich areas. Thus, CXCR3 ligands could mediate T cell trafficking but CAFs could prevent T cells from reaching the malignant cells.

## 1. Introduction

Pancreatic cancer has a 5 year-overall survival rate of less than 10%, which remains the lowest of all common cancers [1]. Pancreatic ductal adenocarcinoma (PDAC) is the most common type of pancreatic cancer and is recognized by a highly fibrotic tumor microenvironment (TME) mainly containing cancer-associated fibroblasts (CAFs). The fibrosis triggered by CAFs surrounds the tumor nests and obstructs the vessels within the tumors which hinders therapy delivery as well as the infiltration of anti-tumor reactive immune cells. Indeed, tumor-infiltrating lymphocytes (TILs) in pancreatic tumors are often excluded from the tumor nests and trapped in the desmoplastic tumor stroma [2,3,4,5]. Strikingly, clonally expanded T cells have been found in PDAC tumors [6,7], suggesting that primed T cells can accumulate in the tumors, but their response may be suppressed by the TME. Therefore, therapies to increase T cell migration to the tumor epithelium could be of great importance.

Besides the physical barrier of the cellular components in the pancreatic TME, soluble factors also play a decisive role in tumor immune escape. The deregulation of some chemokines in the TME can disrupt effector T cells recruitment and thus, prevent immune responses. CXCL12 plays a key role in cancer progression by binding directly to CXCR4 expressed on cancer cells, promoting proliferation and invasiveness [8,9] and indirectly by recruiting tumor immunosuppressive cells [10,11] or by sequestering T cells in the tumor stroma as shown in several PDAC studies [6,12]. The CXCR3 ligands CXCL9 and CXCL10 have been suggested to promote an anticancer immune response in several types of solid tumor [13,14], but their role in pancreatic cancer is still controversial. Qian et al. showed that high levels of CXCL9 and CXCL10 in plasma were correlated with better overall survival [15]. Another study showed a positive correlation between the CXCL10/CXCL9–CXCR3 axis and the infiltration of effector T cells [16]. In contrast, others have shown an association between CXCL10 levels and cancer cell aggressiveness, gemcitabine resistance and worse survival [17,18,19]. The CCR5/CCL5 chemokine axis could also have a dual role in pancreatic cancer. Tan et al. showed that tumor-derived CCL5 recruited T regulatory cells which led to tumor cell growth [20]. However, a recent study by Huffman et al. showed that CCL5 rather promote the infiltration of antitumor CD4^+^ T cells in response to CD40 agonist in a pancreatic mouse model [21].

We and others have previously shown that pancreatic CAFs have immunosuppressive effects on T cells [22,23], but the role of CAFs in modulating chemokine receptors is still largely unknown. Here, we examined how primary pancreatic CAFs modulated the expression of CXCR3, CXCR4 and CCR5 on T cells. Moreover, we investigated the association between the spatial distribution of T cells in tumor tissues and chemokine secretion from pancreatic tumor explants.

## 2. Materials and Methods

### 2.1. Patient Samples

Patients (n = 19) with resectable pancreatic cancer gave written informed consent and were enrolled in the study. The patients underwent surgery at the Upper Gastrointestinal Disease Unit at Karolinska University Hospital, Huddinge, Sweden. CAFs were isolated from 16 out of the 19 patients and expanded to perform in vitro studies. Tissue explants from 10 out of the 19 patients were used for Luminex and immunohistochemistry analyses (Table 1).

Peripheral blood was collected both from patients before surgery and healthy volunteer blood donors. The study and the experimental protocols were approved by the regional ethical review board in Stockholm (entry nos. 2018/1792-31/2, 2019-03500).

### 2.2. Tissue Collection

From the resected tissues, a central part of the tumor, a peripheral part of the tumor and a nontumor-adjacent tissue were collected by a pathologist subspecialized in pancreas pathology at the Pathology Unit at Karolinska University Hospital. A reference tissue sample was also collected for each region, embedded in paraffin, and processed for histology and immunohistochemistry.

### 2.3. Cell Isolation

Primary CAFs were isolated and grown from surgically resected human pancreatic tumor tissues as previously described [3]. Briefly, CAFs were let to migrate out of the tissue fragments and expanded up to passage 3–4 in Dulbecco’s modified Eagle’s medium (DMEM) (Cytiva, Marlborough, MA, USA, cat. no. SH30021.01) supplemented with 10% fetal bovine serum (FBS) (Cytiva, cat. no. SV30160.03) and 1% penicillin–streptomycin (PEST) (Cytiva, cat. no.SV30010) (complete DMEM) under standard culture conditions (37 °C, 5% CO_2_). Expanded CAFs were cryopreserved in complete DMEM with 10% dimethyl sulfoxide (DMSO) (GmbH Cat no. WAK-DMSO-70) in a controlled rate cooling of −1 °C/minute before storage in liquid nitrogen, until use. PBMCs from buffy coats of healthy volunteers or peripheral blood from patients were isolated by a density gradient over Ficoll-Paque PLUS^TM^ gradient (Cytiva, cat. no. 17144003) and cryopreserved in RPMI-1640 (Cytiva, cat. no. SH3035501) supplemented with 10% FBS and 1% PEST (complete RPMI) with 10% DMSO.

### 2.4. Two-Dimensional Cultures

To study the interactions between T cells and CAFs, PBMCs from healthy donors or from patients as indicated were plated in 24-well plates (1 × 10^6^) in the presence or absence of CAFs at a 1:10 ratio (1 CAF per 10 PBMCs) in complete RPMI for 5 days. For some experiments, T cells were stimulated with OKT3 (25 ng/mL) (Biolegend, San Diego, CA, USA, cat. no. 317325) for 5 days.

For a set of experiments, T cells were isolated from PBMCs by negative immunomagnetic selection, using a Pan T cell isolation kit (Miltenyi Biotec, Bergisch Gladbach, Germany, cat. no. 130-096-535) according to the manufacturers protocol. A 12-well Transwell system (0.4 μm pore size membrane (Corning, NY, USA, cat. no. CLS3460)) was used to separate CAFs and PBMCs. CAFs (2 × 10^5^) were added to the upper chamber and PBMCs (2 × 10^6^) were added to the lower chamber.

To study the effects of different factors on migration and chemokine receptor expression on T cells, CXCL10 (Peprotech, Rockhy Hill, NJ, USA, cat. no. 300-12), CXCL12 (Peprotech, cat. no. 300-28B), TGF-β (eBioscience, San Diego, CA, USA, cat. no. 14-8348-62) and PGE_2_ (Sigma Aldrich, St. Louis, MO, USA, cat. no. P0409) were added at indicated concentrations.

Blocking experiments were performed in 96-well plates (2 × 10^4^ CAFs: 2 × 10^5^ PBMCs) and 20 μM indomethacin (Sigma-Aldrich, St. Louis, MO, USA, cat no. I7378) for PGE_2_ blocking, 5 μg/mL anti-TGF-β (R&D systems, Minneapolis, MN, USA, cat no. MAB246), and the appropriate isotype (IgG1 R&D systems, cat no. MAB002, Minneapolis, MN, USA) or vehicle were added to the cell cultures.

For all the 2D experiments, PBMCs were collected on day 5 and analyzed by flow cytometry.

### 2.5. Spheroids

PANC-1 spheroids were established as previously described [24]. Briefly, 2500 PANC-1 cells were mixed in 100 μL of complete DMEM medium supplemented with methylcellulose (Sigma-Aldrich, St. Louis, MO, USA, cat. no. M0512) at a final concentration of 0.24% and cultured in non-tissue-culture-treated 96-well plates. Cells were let to form spheroids for 4 days under standard culture conditions before using them in 3D cultures.

### 2.6. Three-Dimensional Models and Cultures

Collagen matrices were prepared as previously described [24]. Briefly, CAFs (1 × 10^5^), 20 PANC-1 spheroids and PBMCs (1 × 10^6^) were embedded in PureCol bovine type I collagen (Advanced Biomatrix, San Diego, CA, USA, cat. no. 5005) mixed with 5 × DMEM, NaHCO_3_, FBS and PEST to get a final concentration of 2 mg/mL of collagen I.

On day 5, PBMCs were released from the matrices by mechanical disaggregation using a GentleMACS^TM^ Dissociator and analyzed by flow cytometry.

### 2.7. Flow Cytometry

Harvested PBMCs from 2D and 3D cultures were washed and stained with appropriate monoclonal antibodies (Appendix A), in PBS supplemented with 2 mM EDTA and 0.2% bovine serum albumin. Then, 7AAD was used to distinguish live from dead cells. Cells were acquired on a FACSCanto II (BD) or CytoFlex (Beckman Coulter) and data were analyzed using FlowJo (BD) version 10.7.2.

### 2.8. Transwell Migration Assays

Chemotactic responses towards CXCL10, CXCL12 and CCL5 of CD4^+^ and CD8^+^ T cells precultured with CAFs were measured by a transwell migration assay. PBMCs were precultured alone or with CAFs for 5 days in complete RPMI and OKT3 (25 ng/mL) prior to starting the assay, and then 100 μL of cell suspension (5 × 10^5^) in RPMI was added on top of the transwell insert (5 μm pore size (Corning, cat. no. 3425, Corning, NY, USA)). On the bottom of the transwell insert, 600 μL of complete RPMI containing either CXCL10 (20 ng/mL) or CXCL12 (20 ng/mL) was added. A titration for CCL5 (10 ng, 50 mg, 100 ng) was performed. After 2 h of incubation in standard conditions, the inserts were removed and the migrated cells were harvested and stained for CD3, CD4, CD8 and 7AAD (Appendix A). For all the conditions, the cells were resuspended in the same volume of FACS buffer and run on the CytoFlex for the same amount of time and speed.

### 2.9. Tissue-Conditioned Medium and CAFs-Conditioned Medium

A total of 10 paired tissues (central, peripheral and nontumor tissue), were washed once with PBS, cut with a scalpel, and ~50–70 mg of tissue was plated per well in 24-well plates. DMEM medium supplemented with 10% FBS and 1% PEST was added in proportion to the tissue weight (10 μL/mg of tissue). After 48 h incubation at 37 °C, the conditioned medium (CM) of the same tissue area was pooled, centrifuged and frozen in aliquots at −80 °C.

CAFs isolated from 5 PDAC donors were cultured in 6-well plates (2 × 10^5^) in 2 mL of complete DMEM medium and the CM was collected after 48 h.

### 2.10. Multiplex Chemokine Assay and ELISAs

The CM from tissues and CAFs were analyzed for chemokine concentrations using a bioplex array system with the Bio-Plex Pro Human Chemokine 40-plex kit (BIO-RAD Hercules, CA, USA) according to the manufacturer’s instructions. The CM from tissues were diluted 1:2 in complete DMEM and the CM from CAFs were analyzed without dilution. The sensitivity for the chemokines measured were as follows: 1.1 pg/mL for CXCL9, 1.1 pg/mL for CXCL10, 0.05 pg/mL for CXCL11, 15.4 pg/mL for MIF, 10.3 pg/mL for CXCL12, 0.3 pg/mL for CCL3 and 0.04 pg/mL for CCL8. The concentrations of CCL5 in the CM from tissues and CAFs were measured with a CCL5 ELISA (R&D, cat no. DY278-05) (sensitivity 15–1000 pg/mL), according to the manufacturer’s instructions.

### 2.11. Immunohistochemistry Stainings

Consecutive paraffin-embedded sections from the donor tissues were morphologically analyzed by immunohistochemistry analysis. Four-micrometer-thick sections were stained using a Leica BOND III automated immunostainer. The antibody panel consisted of anti-CD4 (clone SP35, Ventana, Oro Valley, AZ, USA), anti-CD8 (clone SP57, Ventana), and anti-CK19 (clone A53-B/A2.26, Cell Marque, Rocklin, CA, USA) antibodies. Anti-CD8 and -CD4 antibodies were combined with anti-CK19 stainings in duplex immunohistochemistry and stained using DAB (brown) and AP (red) chromogens, respectively, to visualize the spatial relation between T cells and tumor cells. A quantitative analysis of T cells and tumor areas was performed on the whole slide images using QuPath (version 0.2.3) [25]. Firstly, pixel classification was used to create annotations for tumor and stroma areas based on positive (red) and negative CK19-staining. The boundary of the tumor annotations was subsequently expanded 20 μm, 50 μm and 100 μm in order to examine the T cell density in stromal areas in close proximity to the tumor nests. A positive cell count was then used to analyze the number of CD4^+^ and CD8^+^ T cells per mm^2^ within all annotations. In cases where the tissues also contained interspersed residual normal tissue, these areas were excluded from the analysis by manual annotation. Quantitative data were exported in a tabular format for the statistical analyses.

### 2.12. Statistical Analysis

A Wilcoxon signed rank test was used to detect differences across two paired groups. Friedman’s test followed by Dunn’s multiple comparison test was used to detect differences between three or more paired groups. A *p*-value < 0.05 was considered statistically significant. All statistical analysis were performed using GraphPad Prism version 9 (La Jolla, CA, USA).

## 3. Results

### 3.1. Pancreatic CAFs Modulate Expression of Chemokine Receptors on T Cells in 2D and 3D Models

To investigate whether pancreatic CAFs modulated chemokine receptor expression on T cells, PBMCs from healthy donors were cultured in the absence or presence of CAFs for 5 days in direct cocultures. Flow cytometry analysis revealed that CAFs downregulated the expression of the Th1 and effector-T-cell-associated receptor CXCR3 on both CD4^+^ and CD8^+^ T cells (Figure 1A,B). In contrast, CAFs upregulated the expression of CXCR4 on CD4^+^ and CD8^+^ T cells, a chemokine receptor that has been suggested to retain T cells in the stroma through CAF-derived CXCL12 secretion [12] (Figure 1A,C). CAFs also reduced the expression of CCR5 on CD8^+^ T cells, but the effect was less consistent between donors (Figure 1A,D). The same trend was observed when patient-derived PBMCs were cultured with autologous CAFs, shown in red connecting lines in Figure 1B–D, thus ruling out that the observed effect was a response to an allogenic reaction. Similar results for CXCR3 and CXCR4 were observed when T cells were stimulated with an anti-CD3 monoclonal antibody, OKT3, suggesting that CAFs can also affect chemokine receptor expression in activated T cells (Appendix A). However, upon activation, CCR5 was only downregulated on CD4^+^ and not CD8^+^ T cells by CAFs.

We have previously shown that plastic adherence in 2D cultures promote CAF activation and the upregulation of alpha-smooth muscle actin (α-SMA) expression, a hallmark of myofibroblastic CAFs [24]. To examine if the altered chemokine expression was an effect of the culture conditions, we next examined the modulatory capacity of CAFs on T cells in 3D collagen matrix models to better mimic a tissue microenvironment. Figure 1B–D show that CAFs embedded in collagen matrix in 3D cultures mediated similar effects on the expression of chemokine receptors in T cells as in 2D cultures. Together, this suggests that CAFs promote the expression of a chemokine receptor involved in preventing T cell access to tumor cells and decrease the expression of chemokine receptors previously described to stimulate the migration towards malignant cells.

To examine if a direct cell-to-cell contact between T cells and CAFs was necessary for affecting chemokine receptor expression, we used transwell assays where the CAFs were separated from the immune cells. The expression of CXCR3 was increased and CXCR4 was decreased on T cells when CAFs were separated by a transwell, and there was no difference in the expression of CXCR3 and CXCR4 between T cells in direct contact with CAFs or in transwells, suggesting that the effect is mainly mediated by soluble factors (Figure 1E–H). However, CAFs had a weaker but significant effect on CCR5 expression when separated by a transwell membrane (Figure 1E–H).

PBMCs are a heterogenous population of cells, which include monocytes and lymphocytes. It is known that CAFs can redirect M1 macrophages to M2 macrophages [26] and can induce T regulatory cells [3]. To investigate whether the effect of CAFs on CXCR3 and CXCR4 expression on T cells was a direct effect or an indirect effect through crosstalk with other cell types, isolated T cells and PBMCs were cultured with or without CAFs for 5 days. Appendix A shows that CAFs downregulated CXCR3 and upregulated CXCR4 on CD4^+^ and CD8^+^ T cells independently of monocytes or other cell types.

To investigate if pancreatic tumor cells could also affect chemokine receptor expression on T cells, PBMCs were cocultured with PANC-1 spheroids in the presence or absence of CAFs in collagen matrices. No significant effect on the expression of CXCR3 and CXCR4 was observed neither on CD4^+^ nor on CD8^+^ T cells when PBMCs were cultured in the presence of tumor spheroids compared to PBMCs alone (Figure 1I–K). However, in line with the observations above, when CAFs were added to the cultures together with tumor spheroids, a decrease in CXCR3 expression and an increase in CXCR4 expression was observed in CD8^+^ T cells. Tumor spheroids increased the expression of CCR5 on CD4^+^ T cells, but this significance was lost when both CAFs and tumor spheroids were added. This suggests that CAFs are counteracting the effect of the tumor spheroids by reducing the CCR5 expression (Figure 1I,L). To summarize, these data show consistent effects of CAFs on the downregulation of CXCR3 and the upregulation of CXCR4 expression on T cells in both 2D and 3D cultures, as well as when tumor spheroids were present in the 3D cultures.

### 3.2. CAF-Released PGE_2_ Partially Modulate CXCR4 Expression on CD8^+^ T Cells

Factors involved in the regulation of the surface expression of chemokine receptors are not well established, but chemokines could influence the expression of their corresponding chemokine receptors [27]. To map the chemokine secretion pattern from CAFs, we analyzed the secretion of CXCR3, CXCR4 and CCR5 ligands from pancreatic CAFs. We found that the CXCR3 ligands CXCL10, CXCL9 and CXCL11, and the CCR5 ligands CCL3 and CCL8 were modestly produced by CAFs (Figure 2A,B). CCL5 was under the detection level of the ELISA (Figure 2C). However, in line with other studies [12], we observed that the CXCR4 ligand CXCL12 was produced in high levels. Macrophage migration inhibitory factor (MIF), a cytokine with migratory function through CXCR4 signaling, was also excessively produced by CAFs (Figure 2C). To investigate if CXCL10 and CXCL12 levels produced by CAFs could affect the expression of CXCR3 and CXCR4 on T cells, we added different concentrations of recombinant CXCL10 and CXCL12 to PBMCs, covering the CAF-secreted range. Three independent experiments showed no clear effect of CXCL10 or CXCL12 on the CXCR3 and CXCR4 expression on T cells (Figure 2D,E).

Next, we investigated whether the immunosuppressive molecules TGF-β, and PGE_2_, known to be secreted by CAFs [24,28], could alter the chemokine receptor expression on T cells. We found that the recombinant TGF-β upregulated CXCR3 on CD8^+^ T and CXCR4 on both CD4^+^ and CD8^+^ T cells at a concentration of 1 ng/mL (Figure 2F). On the other hand, PGE_2_ did not alter the expression of CXCR3 (Figure 2G) but upregulated the expression of CXCR4 on CD4^+^ T cells at 50ng/mL and on CD8^+^ T cells at 30 ng/mL (Figure 2G). By blocking CAF-derived TGF-β signaling using anti-TGF-β-neutralizing antibodies the chemokine receptor expression was not restored (Figure 2H). However, the blockade of CAF-derived PGE_2_ with indomethacin led to a partially decreased expression of CXCR4 on CD8^+^ T cells (Figure 2I). This indicates that PGE_2_ is partially involved in the regulation of CXCR4 expression but that other CAF-derived soluble factors are also involved.

### 3.3. CAFs Modulate T Cell Migratory Ability

To examine if the CAF-mediated regulation of chemokine receptor expression resulted in functional alterations in the migratory capacity of activated T cells, we next investigated how T cells precultured with CAFs for 5 days migrated towards CXCL10 (20 ng/mL) and CXCL12 (20 ng/mL). The amount of CXCL10 and CXCL12 was decided after a titration (Appendix A). T cell chemotaxis towards CXCL10 was strongly inhibited after being cultured with CAFs with a more pronounced effect on CD8^+^ T cells. No significant difference was seen on the migratory ability of T cells towards CXCL12 (Figure 3). In line with other studies [29], CCL5 did not attract T cells in our in vitro model (Appendix A).

### 3.4. Spatial Distribution of T Cells within the Desmoplastic Tumor and Association with Chemokine Production

We next aimed to investigate if chemokine secretion from pancreatic tumor tissues affected T cell infiltration and their spatial localization within the desmoplastic tumor. First, we studied whether there was a chemokine gradient towards tumor rich areas. Three different tissues were resected from the same PDAC patient: a central tumor tissue, a peripheral tissue (rich in tumor) and a nontumor tissue (tumor-free area). The tissues were cultured for 48 h and the protein levels of the CXCR3, CXCR4 and CCR5 ligands were measured by Luminex or ELISA. We found that the secretion of CXCL9, CXCL10 and CXCL11 was significantly higher in central tumor tissues compared to nontumor tissues (Figure 4A). However, the CXCR4 and CCR5 ligands were more uniformly expressed in tumor and nontumor tissues (Figure 4B,C).

Paired tumor tissues were stained immunohistochemically to study the number and location of CD4^+^ and CD8^+^ T cells in central and peripheral tumor tissues and quantified using QuPath. A Pixel classification was used to create annotations for tumor areas based on CK19-staining and stroma on unstained tissue (Figure 4D). To investigate T cell distribution within areas at different distances from the tumor nests, the exterior of the tumor annotations was expanded 20 μm, 50 μm and 100 μm from the tumor. A positive cell count was used to analyze the number of CD4^+^ and CD8^+^ T cells per mm^2^ within all annotations. Since both peripheral and central desmoplastic tumor tissues were infiltrated by similar numbers of T cells (Figure 4E), we pooled data from both compartments in the subsequent analysis. T cells were mainly found in the tumor stroma with very few cells infiltrating tumor epithelial nests with a median of 19 and 11 cells/mm^2^ for CD4^+^ and CD8^+^ T cells, respectively (Figure 4F). CD4^+^ T cells were equally distributed throughout the stroma but significantly less present in the tumor nests. However, CD8^+^ T cells were significantly fewer in areas 20 μm from the tumors compared to within the total stroma, suggesting that cytotoxic T cells to a larger degree were excluded from the malignant cells. In line with this, the ratio between CD4^+^ and CD8^+^ T cells was significantly higher in the tumor nest and 20 μm and 50 μm from the tumor compared to the total stroma (Figure 4G).

We next examined if CD8^+^ T cell infiltration into malignant tissue was associated with levels of CXCR3-, CXCR4- and CCR5 ligand secretion. The tissues were divided into high and low CD8^+^ T cell infiltration based on the median CD8^+^ T cell count per mm^2^ (Figure 5A). Surprisingly, we found that tissues with a high CD8^+^ T cell infiltration in the total stroma had a significantly lower secretion of most chemokines than tissues with a low presence of CD8^+^ T cells (Figure 5B). Levels of CXCL10, CXCL11, CXCL12 and CCL8 also showed a negative correlation with the number of CD8^+^ T cells in the total stroma (Figure 5C and Appendix A). Tumor nests contained too few CD8^+^ T cells to perform a reliable analysis, but assuming that CD8^+^ T cells localized within the tumor and within 20 μm from the tumor could mediate cytotoxic killing of malignant cells, we combined the tumor and the stromal area 20 μm from the tumor into one annotation to analyze the number of CD8^+^ T cells with putative capacity to interact with tumor cells (Figure 5D). As expected, CD8^+^ T cells within this area were significantly fewer than in the total stroma (Figure 5A). However, in contrast to the total stroma, we found no significant difference in chemokine production between tissues with a high and low CD8^+^ T cell count (Figure 5E) and there was no correlation to the CD8^+^ T cell count (Figure 5C). To investigate if the relative localization of CD8^+^ T cells within the desmoplastic tumor was associated with a chemokine secretion pattern, we calculated the ratio between the number of CD8^+^ T cells in areas within reach of the tumor cells and the total stroma (Figure 5F). Interestingly, tissues with a high CD8^+^ T cell ratio had higher secretion levels of CXCL10, CXCL11 and CCL8 chemokine levels than tissues with a low CD8^+^ T cell ratio (Figure 5G). Furthermore, we found that CXCL9, CXCL10, CXCL11 and CCL8 levels were positively correlated with a higher ratio between CD8^+^ T cells in tumoral areas and the total stroma (Figure 5C,H).

Similar analyses were performed for CD4^+^ T cells (Appendix A). As mentioned above, CD4^+^ T cells were more equally distributed within the stromal tissues and no significant difference was found between the number of T cells in the total stroma and T cells in tumoral areas (tumor nests + 20 μm from tumor) (Appendix A). Tissues from patients with a higher accumulation of CD4^+^ T cells in the total stroma secreted less CXCL12 (Appendix A). However, as with CD8^+^ T cells, Spearman analyses indicated a negative correlation between the number of CD4^+^ T cells in the total stroma and CXCL10, CXCL11, CXCL12 and CCL8 levels (Appendix A). Tissues from patients with low numbers of CD4^+^ T cells within 20 μm of the tumor secreted less MIF (Appendix A) and there was a negative correlation between MIF levels and numbers of CD4^+^ T cells within and 20 μm of the tumor nests (Appendix A).

Finally, unlike CD8^+^ T cells, no significant differences were found between the levels of chemokine secretion and patients with high or low ratio of CD4^+^ T cells within and close to the tumor nest (Appendix A). However, Spearman analyses showed a positive correlation between levels of CXCL10, CXCL11 and CCL8 and a high CD4^+^ T cell ratio (Appendix A).

To summarize, these data suggest that a high general secretion of chemokines is associated with a poor T cell infiltration into the tumoral desmoplastic stroma, but that CXCR3 ligands and the CCR5 ligand CCL8 correlate with a high CD8^+^ and CD4^+^ T cell infiltration into areas in proximity to malignant cells in relation to the total desmoplastic stroma.

## 4. Discussion

Over the last two decades, efforts have been focused on targeting the pancreatic TME due to its decisive role in tumorigenesis and immune evasion. The urge to identify and understand the influencing factors is important to develop novel treatment strategies and overcome the immunosuppressive TME. The infiltration of CD8^+^ T cells in the pancreatic tumors has been correlated with a better prognosis [4,30]. However, little is known about the regulatory molecules that alter the transit of the effector T cells in the pancreatic TME. In this in vitro study, we showed that primary pancreatic CAFs can potentially influence T cell trafficking by upregulating CXCR4 and downregulating CXCR3 on T cells. Our data also suggest that the ligands for the chemokine receptors CXCR3 (CXCL9, CXCL10, CXCL11) and CCR5 (CCL8) might affect the spatial distribution of CD8^+^ T cells in human pancreatic tumor tissues and enhance the relative T cell trafficking into tumor rich areas.

In a recent study, Vonderhaar et al. showed that PDAC mice models treated with a stimulator of interferon genes agonist (STING) had increased levels of the proinflammatory chemokines CXCL9/CXCL10 which prompted the infiltration of effector T cells in a CXCR3-dependent manner [16]. CD8^+^ T cells recruitment through CXCR3 signaling have also been reported in melanoma and other solid tumors [13,31]. In the present study, we observed that CAFs consistently decreased the expression of CXCR3 on T cells and that this led to a diminished capacity to migrate towards CXCL10 in vitro. This suggests that CAFs may contribute to the entrapment of T cells in the desmoplastic stroma by disabling CXCR3 ligation and hence a migration towards the tumor nests. We further found that tumor-rich tissues produced higher levels of CXCL9, CXCL10 and CXCL11 compared to nontumor tissues. Moreover, tumor tissues with a high T cell ratio within and close to the tumor nests in relation to the total stroma produced higher levels of CXCL9, CXCL10 and CXCL11, signifying the importance of the CXCR3-axis in T cell localization within the tumor microenvironment.

Paradoxically, we further found that a high secretion of CXCR3 ligands was associated with a poorer T cell density within the total tumor and stromal tissue when the spatial localization of T cells was unaccounted for. This is in line with several other studies in PDAC that showed a correlation between a higher expression of CXCR3 ligands and a poor prognosis based on mRNA expression [17,18,19]. However, our study suggests that a high secretion of chemokines in general is associated with fewer T cells within the desmoplastic tumor since this observation was not confined to CXCR3 ligands, but also to the CXCR4 and CCR5 ligands. It can be speculated that tumors with a poor T cell infiltration are associated with chronic inflammation with a high infiltration of chemokine-producing suppressive myeloid cells. Myeloid-derived suppressor cells (MDSC) have been shown to be prevalent in PDAC and promote T cell exclusion [32]. The controversial results on the role of CXCR3 ligands in pancreatic cancer could also be explained by the fact that a subset of cancer cells in pancreatic cancer express CXCR3 and that the exposure to CXCL10 promotes resistance to gemcitabine [19]. Thus, CXCR3 ligands can have different roles depending on the target. In the present study, we did not evaluate the role of the CXCR3 ligands on tumor cells and it remains to be determined which cells produce these proinflammatory chemokines in the TME. Altogether, our results suggest a role of the CXCR3 ligands CXCL9, CXCL10 and CXCL11 in T cell trafficking and that CAFs could jeopardize T cell mobility. However, further studies with a larger number of samples are needed to confirm these findings.

The CXCL12–CXCR4 axis has been shown to be correlated with a poor prognosis in pancreatic cancer and to enhance cancer cell proliferation and survival [9]. Besides its role on cancer cells, Feig et al. also showed in a PDAC murine model that CXCL12 produced by CAFs could retain T cells through CXCR4 signaling in the tumor stroma [12]. Targeting CXCR4 induces the mobilization of CD8^+^ T cells to the tumor nests and enhances the clinical response to immunotherapy in pancreatic cancer [6,12,32]. We showed that primary CAFs isolated from PDAC patients secreted the ligands for CXCR4, CXCL12 and MIF, and that CAFs consistently upregulated CXCR4 expression on CD4^+^ and CD8^+^ T cells in both 2D and 3D cocultures. This suggests that CAFs have dual roles in retaining T cells in the CAF-rich stroma in the context of the CXCR4 axis, both by providing CXCR4 ligands and by increasing the expression of CXCR4. However, in our in vitro settings, T cells precultured with CAFs did not significantly increase the migration towards a CXCL12 gradient. This could potentially be because the CXCR4 expression on T cells was relatively high at the steady state and that the CAF-mediated upregulation of CXCR4 was not sufficient to increase the migratory capacity in vitro.

A recent study has shown that CAF-derived TGF-β is associated with T cell exclusion in metastatic urothelial cancers [33]. Among other mechanisms, this could be mediated by the upregulation of CXCR4 on T cells [34,35] and by inhibiting IFN-γ signaling in T cells [36], which could lead to a decreased expression of CXCR3 on T cells [37] and thus, prevent T cell trafficking [13]. Consistent with other studies, we showed that CXCR4 was upregulated by TGF-β on both CD4^+^ and CD8^+^ T cells. However, the blockade of TGF-β did not restore the levels of CXCR4 on T cells in CAF cocultures. We have previously shown that primary CAFs release high levels of PGE_2_ [24], which exert regulatory functions on T cells [3]. PGE_2_ has also been shown to upregulate the expression of the CXCL12–CXCR4 axis and to enhance stromal formation, angiogenesis [10] and the recruitment of MDSC [11]. Other in vitro studies have shown a regulatory function of PGE_2_ on the CXCR3–CXCL9/CXCL10 axis, by impairing the release of CXCL9 and CXCL10 [38] and downregulating CXCR3 expression on T cells [39]. In our in vitro settings we did not observe any effect of PGE_2_ on CXCR3 expression but high levels of PGE_2_ induced the expression of CXCR4. Moreover, the blockade of CAF-derived PGE_2_ with indomethacin partially restored the expression of CXCR4 on CD8^+^ T cells. To summarize, our data provide further evidence of the upregulation of CXCR4 on T cells by CAFs and that it can partially be mediated by PGE_2_ secretion. However, other factors not identified by us are likely involved in the CAF-mediated effects on the expression of chemokine receptors.

In this study we also showed that CAFs suppressed CCR5 expression on CD8^+^ T cells but not on CD4^+^ T cells, but the effect was not consistent in all the donors. Interestingly, the tumor spheroids in 3D cultures upregulated CCR5 expression on CD4^+^ T cells, but the effect was counterbalanced by CAFs. Moreover, when T cells were activated by OKT3, CAFs significantly downregulated the expression of CCR5 on CD4^+^ T cells. This indicated that CAFs could also influence CD4^+^ T cells and thus, their migration. However, the role of the CCR5 ligands CCL3, CCL5 and CCL8 in pancreatic cancer remains controversial. Several studies have shown that CCL5 and CCL8 promote cancer cell proliferation and migration [40,41] as well as the recruitment of T regulatory cells, which leads to tumor progression [20]. In contrast, a recent study suggested that MDSC-derived CCL5 promoted the infiltration of antitumor CD4^+^ T cells [21]. We were not able to study whether T cells precultured with CAFs had an affected chemotaxis towards a CCL5 gradient since, as published by others, the CCL5 chemotactic activity on T cells in vitro is weak [29]. However, we showed that the spatial distribution of CD8^+^ T cells, but not of CD4^+^ T cells, could be affected by the CCR5 ligand CCL8. A high ratio of CD8^+^ T cells within the tumor areas in relation to the total T cells in the stroma was correlated with CCL8 secretion. However, further studies with a larger number of samples are necessary to confirm these findings. Importantly, previous studies in lung, ovarian and prostate cancers have shown elevated levels of CCL5 secreted by CAFs, which led to therapy resistance and cancer cell survival [42,43,44]. However, in our in vitro models neither cultured CAFs (nondetectable levels) nor tissue explants, which are rich in CAFs, released high levels of CCL5 (median 120 pg/mL).

A recent immunohistochemistry and RNA-sequencing meta-analysis of resectable and advanced PDAC specimens showed a positive correlation between CCL5, CXCL9 and CXCL10 and CD8^+^ T cells infiltration in close proximity to tumor cells. Moreover, the same study showed that a high coexpression of the four chemokines CCL4, CCL5, CXCL9, and CXCL10 was associated with the infiltration of cytotoxic CD8^+^ T cells [45]. In our study, we did not find any correlation between chemokine secretion and the total number of T cell infiltration in the vicinity or in the tumor nests. However, when we took into account the relative localization of T cells within the tumor, we also found a positive correlation between CXCL9, CXCL10 and a high CD8^+^ T cell ratio within and close to tumor nests.

Our immunohistochemistry stainings and analysis further showed that the number of CD8^+^ T cells was lower in areas close to the tumor nests compared to the total stroma. No such difference in distribution within the stroma was observed for CD4^+^ T cells, suggesting that CD4^+^ T cells to a larger degree than CD8^+^ T cells migrate towards malignant cells. Regulatory T cells constitute a significant proportion of all CD4^+^ T cells in PDAC tumors [46], and it can be speculated that regulatory T cells have superior capacities to migrate towards malignant cells compared to conventional T cells [20].

A limitation of our study on the spatial distribution of T cells is the relatively low number of specimens. This highlights the difficulties of obtaining such material but also manifests the value of our results. Our pilot study has provided clues to the complex communication between T cells and chemokines within the pancreatic TME, but further studies with a larger sample size are required. Further studies are also warranted including more chemokines as well as the T cell clonality and functional capacity. It would also be of great interest to study the role of the different CAF subsets on the modulation of chemokine receptors which would help to identify which chemokines influence antitumor T cell trafficking and develop new therapies for pancreatic cancer.

To summarize, our results suggest that certain CAF-derived soluble factors downregulated the expression of CXCR3 on CD4^+^ and CD8^+^ T cells and CCR5 on CD8^+^ T cells and upregulated the expression of CXCR4 on CD4^+^ and CD8^+^ T cells, which affect their migratory capacity towards tumor cells (Figure 6A). The previously described importance of the CXCR4–CXCL12 axis for T cell entrapment in the desmoplastic stroma may not only be due to CAF-secreted CXCL12, but also to an upregulation of its receptor. The CAF-mediated reduction in CXCR3 expression on T cells could incapacitate an efficient migration towards tumor nests, which may be of particular importance in PDAC due to its high content of CAF-rich stroma. Our immunohistochemistry analyses showed that tumor tissues with a high secretion of chemokines contained fewer numbers of T cells in the total stroma; however, the CAF-mediated retention of T cells in the stroma could be overcome in the presence of high levels of CXCL10, CXCL11 and CCL8, since the spatial localization of CD4^+^ and CD8^+^ T cells within and close to the tumor nests was correlated with a high secretion of those chemokines (Figure 6B,C). However, it remains to be determined which cells in the tumor microenvironment secrete those chemokines. Our data underscore the importance of chemokine axes in T cell trafficking in PDAC patients, which could provide new strategies for pancreatic cancer immunotherapy.

## 5. Conclusions

In conclusion, our immunohistochemistry results showed that CD4^+^ T cells were equally distributed throughout the stroma, whereas CD8^+^ T cells were localized more distantly from the tumor nests. However, a high ratio of CD8^+^ T cells within the tumor areas in relation to the total T cells in the stroma was correlated with the secretion of CXCR3 ligands (CXCL9, CXCL10 and CXCL11) and the CCR5 ligand CCL8. We also showed that CAF-derived soluble factors downregulated the expression of CXCR3 on CD4^+^ and CD8^+^ T cells and CCR5 on CD8^+^ T cells and upregulated the expression of CXCR4 on CD4^+^ and CD8^+^ T cells, potentially affecting their migratory capacity towards tumor cells.

## Figures and Tables

**Figure 1 cancers-14-03826-f001:**
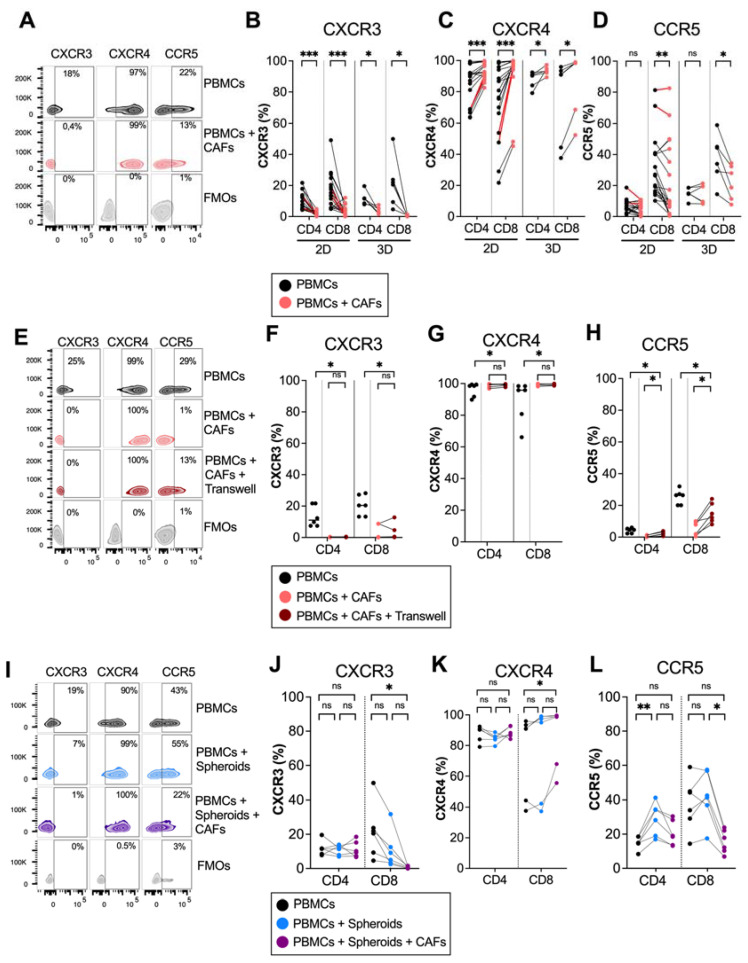
CAFs modulate chemokine receptor expression on T cells in 2D and 3D models. PBMCs were cocultured in the absence (black dots) or presence of CAFs (light red dots) in direct cocultures or in indirect transwell cocultures for 5 days. (**A**) Representative zebra plots showing the gating strategy in (**B**–**D**). (**B**–**D**) Frequency of (**B**) CXCR3, (**C**) CXCR4 and (**D**) CCR5 expression in CD4^+^ and CD8^+^ T cells in 2D (n = 21) and 3D models (n = 6)**.** (**E**) Representative zebra plots showing the gating strategy in (**F**–**H**). (**F**–**H**) Frequency of (**F**) CXCR3, (**G**) CXCR4 and (**H**) CCR5 expression in CD4^+^ and CD8^+^ T cells in direct cocultures (light red dots) or indirect transwell cultures (dark red dots) (n = 6). (**I**) Representative zebra plots showing the gating strategy in (**J**–**L**). (**J**–**L**) Frequency of (**J**) CXCR3 (**K**) CXCR4 and (**L**) CCR5 expression in CD4^+^ and CD8^+^ T cells in 3D models in the absence of spheroids and CAFs (black dots), in the presence of spheroids (blue dots) or the presence of spheroids and CAFs (purple dots) (n = 6). Dots and lines show paired samples. (**B**–**H**) Wilcoxon signed-rank test was used to detect statistically significant differences between paired samples. (**J–L**) Friedman’s test was used to detect statistically significant differences. ns, not significant. * *p*  <  0.05, ** *p*  <  0.01, *** *p*  <  0.001.

**Figure 2 cancers-14-03826-f002:**
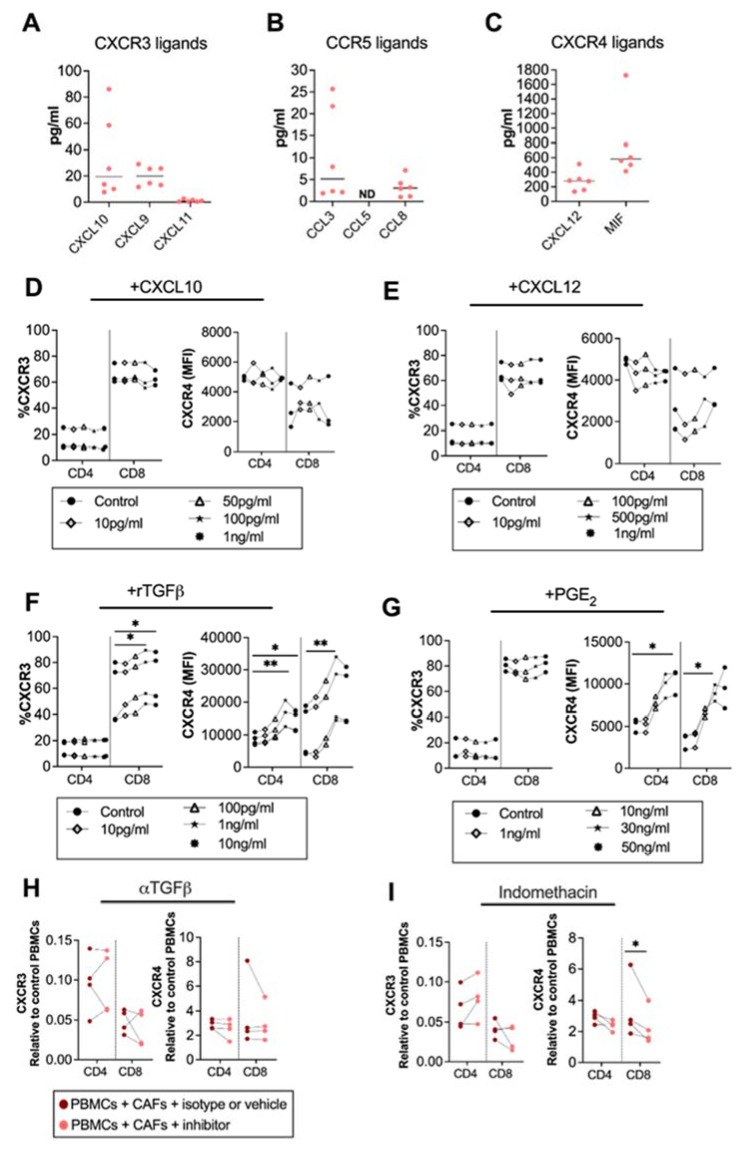
CAF-released CXCL10, CXCL12 and TGF-β are not involved in CAF-mediated modulation of chemokine receptors, but PGE_2_ partially modulates CXCR4. Levels of (**A**) CXCL10, CXCL9 and CXCL11; (**B**) CCL5, CCL3, and CCL8; and (**C**) CXCL12, and MIF in CAF-conditioned medium (n = 6). (**D**–**G**) Frequencies of (**left**) CXCR3 and (**right**) CXCR4 expression in CD4^+^ and CD8^+^ T cells after adding different concentrations of (**D**) CXCL10 (10 pg/mL, 50 pg/mL, 100 pg/mL, 1 ng/mL); (**E**) CXCL12 (10 pg/mL, 100 pg/mL, 500 pg/mL, 1 ng/mL); (**F**) rTGFβ (10 pg/mL, 100 pg/mL, 1 ng/mL, 10 ng/mL); and (**G**) rPGE_2_ (1 ng/mL, 10 ng/mL, 30 ng/mL, 50 ng/mL) to the PBMCs. (**H**,**I**) Frequencies of (**right**) CXCR3, (**left**) CXCR4 on CD4^+^ and CD8^+^ T cells cultured with CAFs after addition of the corresponding isotype, vehicle or the following blocking: (**H**) 5 μg/mL of anti-TGFβ to block TGF-β and (**I**) 20 μM of indomethacin to block PGE_2_. Data are related to those of the control cultures without CAFs. (**D**–**I**) Dots and lines show paired samples. (**D**–**G**) Friedman’s test and (**H**,**I**) a paired *t* test were used to detect statistically significant differences * *p*  <  0.05, ** *p*  <  0.01.

**Figure 3 cancers-14-03826-f003:**
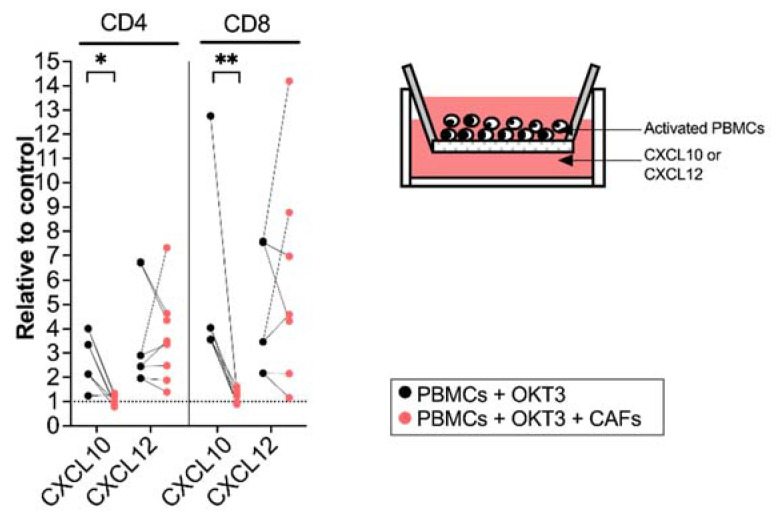
CAFs inhibit T cell migration towards CXCL10. PBMCs were precultured in the absence (black dots) or in the presence of CAFs (light red dots) and activated with OKT3 (25 ng/mL) for 5 days. (**Left**) Number of migrated CD4^+^ and CD8^+^ T cells towards CXCL10 (20 ng/mL) and CXCL12 (20 ng/mL) relative to medium control. (**Right**) Schematic representation of the transwell migration assay. Wilcoxon signed-rank test was used to detect statistically significant differences between paired samples. * *p*  <  0.05, ** *p*  <  0.01.

**Figure 4 cancers-14-03826-f004:**
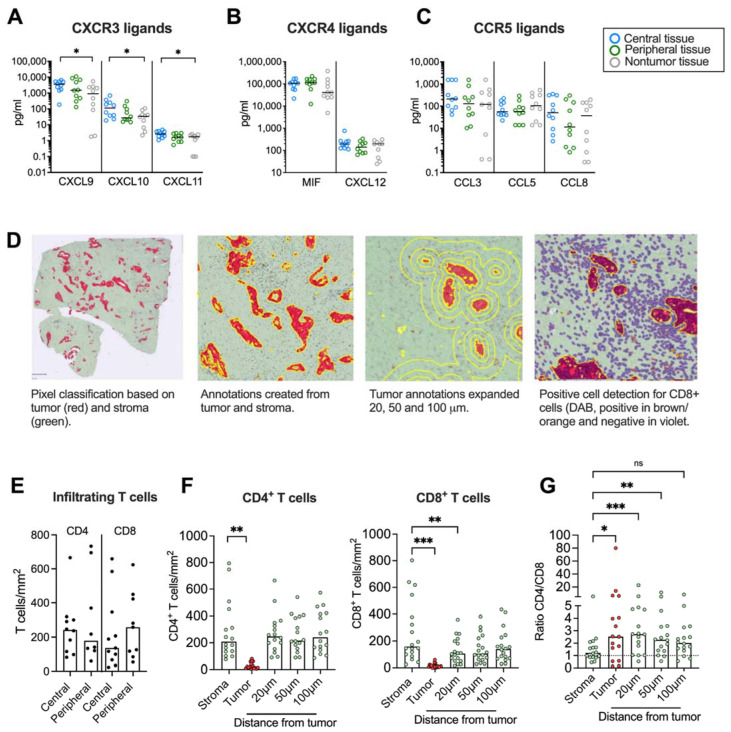
Immunohistochemical analyses of CD4^+^ and CD8^+^ T cells in tumor tissues. (**A**–**C**) Chemokine levels in conditioned medium from tumor tissues cultured for 48h. (**D**, **left**) Pixel classification based on tumor (CK19 in red) and stroma (green). (**Middle left**) Annotations to determine tumor and stroma. (**Middle right**) Annotations to show expanded tumor areas of 20, 50 and 100 μm. (**Right**) Detection of CD8^+^ T cells in brown/orange. (**E**) Infiltrating CD4 and CD8 T cells in central and peripheral tumor tissues (**F**) Number of CD4^+^ and CD8^+^ T cells per mm^2^ in total stroma, tumor nests (red), and 20, 50 and 100 μm from the tumor nest. (**G**) Ratio of CD4^+^/CD8^+^ T cells in total stroma, tumor nests (red), and 20, 50 and 100 μm from the tumor nest. (**A**–**C**,**E**) Wilcoxon signed-rank test was used to detect statistically significant differences between paired samples. (**F**) Friedman’s test was used to detect statistically significant differences. ns, not significant. * *p*  <  0.05, ** *p*  <  0.01, *** *p * <  0.001.

**Figure 5 cancers-14-03826-f005:**
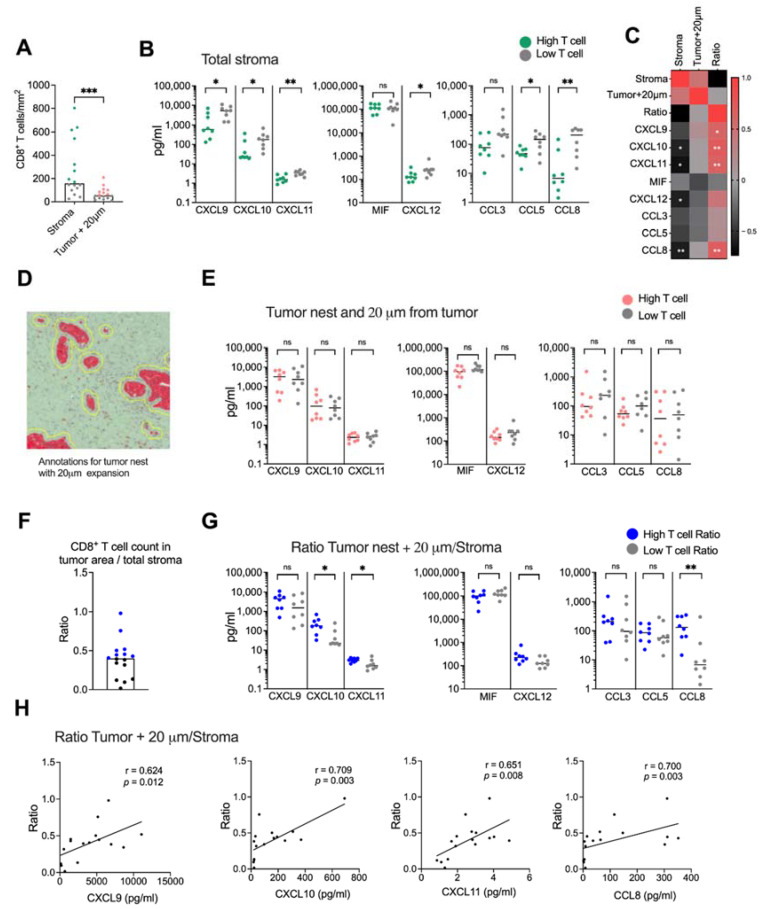
Spatial localization of CD8^+^ T cells. (**A**) Number of CD8^+^ T cells in stroma and in the tumor nest with 20 μm expansion from the tumor nests. Green and red dots show the donors with CD8^+^ T cells numbers above the median and gray dots show the donors with CD8^+^ T cell numbers below the median. (**B**) Secretion of chemokines in tissues with high CD8^+^ T cell numbers (green) and low CD8^+^ T cell numbers (gray). (**C**) Heatmap correlation matrix showing positive (red) and negative (black) correlations between number of CD8^+^ T cells and chemokine secretion. (**D**) Annotations including both the tumor nests and 20 μm from the tumor nests. (**E**) Secretion of chemokines in tissues with high CD8^+^ T cells numbers (red) and low CD8^+^ T cell numbers (gray) in the tumor nests + 20 μm from the tumor nests. (**F**) Ratio between the number of CD8^+^ T cells in tumor nest + 20 μm and the CD8^+^ T cell count in total stroma. (**G**) Secretion of chemokines in tissues with high CD8^+^ T cells ratio (blue) and low CD8^+^ T cell ratio (gray). (**H**) Correlations between the ratio of CD8^+^ T cells in tumoral areas (tumor nest + 20 μm from tumor/total stroma) and CXCL9, CXCL10, CXCL11 and CCL8. Mann–Whitney test was used to detect statistically significant differences between unpaired samples. Correlations were evaluated by using Spearman’s correlation test. Spearman *r* and *p*-values are presented. ns, not significant. * *p*  <  0.05, ** *p*  <  0.01, *** *p*  <  0.001.

**Figure 6 cancers-14-03826-f006:**
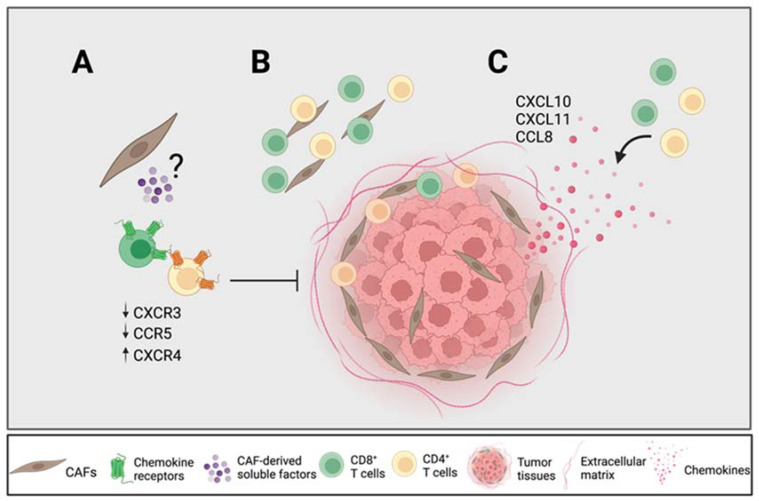
CAFs modulate chemokine receptor expression, which can potentially promote T cell exclusion. (**A**) CAF-derived soluble factors downregulate the chemokine receptors CXCR3 and CCR5 on T cells known to be important for T cell trafficking towards tumor nests and upregulate CXCR4 known to be involved in T cell exclusion. (**B**) CD4^+^ T cells are equally distributed throughout the stroma whereas CD8^+^ T cells are found more distantly from the tumor nests. (**C**) High levels of CXCL10, CXCL11 and CCL8 from tumor tissues are associated with a higher CD4^+^ and CD8^+^ T cell infiltration within and close to the tumor nests in relation to the total stroma. Figure created with BioRender.com.

**Table 1 cancers-14-03826-t001:** Patient characteristics for patients included in the Luminex and immunohistochemistry analysis.

Variables	n = 10
Demographic characteristic	
Female gender, n (%)	7 (70)
Mean age, years (±SD)	60.9 (9.1)
Median age, years (range)	62 (47–72)
Male gender, n (%)	3 (30)
Mean age, years (±SD)	75.3 (3.2)
Median age, years (range)	74 (73–79)
Oncologic characteristics	
Histological type	
Pancreatic ductal adenocarcinoma, n (%)	10 (100)
Tumor depth, n (%)	
T3	8 (66.6)
Lymph node metastasis, n (%)	
N0	8 (80)
N1	2 (20)
Metastasis, n (%)	
M0	8 (80)
M1	2 (20)
Preoperative chemotherapy, n (%)	
Yes	1 (10)
No	9 (90)

## Data Availability

The data presented in this study are available on request from the corresponding author.

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
