# Peer review of "Chemokine Receptor Expression on T Cells Is Modulated by CAFs and Chemokines Affect the Spatial Distribution of T Cells in Pancreatic Tumors"

_cancers, 2022, doi:10.3390/cancers14153826_

Round 1

Reviewer 1 Report

Reviewer comments:

Comments to the Author

The authors have Understanding the mechanisms by which T cells are prevented from reaching the tumor nests is of great importance for the development of novel targeting therapies.

This manuscript is written very well, the experimental progression was logical, and the data provided was comprehensive, well validated and presented clearly along with the discussion including the recent studies.

Author Response

Thank you for your comments.

Reviewer 2 Report

The manuscript by Dr. Laia Gorchs and colleagues examined the idea that pancreatic cancer associated fibroblasts (CAF) can impact the expression of chemokine receptors on T cells in vivo, and hence T cell access to tumor environments and spatial distribution within tumor. This investigation could be of interest to the field especially regarding PDAC and TME of other GI cancers. However, I find the much of the data are weak because of lack of sufficient replicates and/or lack of statistical analysis in the data sets. Thus, conclusions drawn cannot be supported by data as it is presented (lacking asterisks to identify a difference that is significant).

Suggestions/comments.

  1. Figure 1 and section 3.1 text lines 205-258. Statistical analysis in multiple panels is incomplete, or in many cases, was not statistically significant. These data as presented do not support the authors conclusion that pancreatic CAFs modulate chemokine receptor expression in 2 or 3D culture conditions.
  2. Figure 2. Data in panels Fig 2 E – G lack statistical analysis, thus no possible make conclude that TGF-beta or PGE2 impact T cell secretion of chemokines.
  3. Figure 3 and lines 315-319: panels for CD4 and CD8 migration to CXCL10 appear significantly impaired but are missing asterisks that indicate significance.
  4. Fig 4A. AGAIN, although Authors state that CCR3-ligand proteins are higher in central tissue compared to non- tumor tissue, the figure does not indicate statistical significance. Authors need to be more specific and include identifying data that are significant using asterisks – Authors maybe correct that production of CXCL9 and CXCL-10, but not CXCL11, are elevated in central tumor vs non-tumor.

 4B. Same criticisms as above for panels 4E -G, data stated as significantly different was not identified in these panels.

  1. Same criticisms as in #4 for Data in Fig 5.

Author Response

Thanks for all the valid comments. Unfortunately, an error ocurred when the manuscript was uploaded and all asterisks depicting significances were lost, sorry for the inconvenience. We have now uploaded a new version of the manuscript and hope that all significant differences are clearly shown in every figure. We therefore believe that we have addressed all the comments.